# Soil-Dwelling Arthropods' Response to Land Abandonment Is Taxon-Specific in a Mediterranean Olive Grove Agroecosystem

Matteo Dellapiana [1,*], Alice Caselli [1], Gaia Monteforti [2], Ruggero Petacchi [1] and Anna-Camilla Moonen [1]

1 Institute of Plant Sciences, Sant'Anna School of Advanced Studies, Piazza Martiri della Libertà 33, 56127 Pisa, Italy; alice.caselli18@gmail.com (A.C.); ruggeropetacchi@gmail.com (R.P.); camilla.moonen@santannapisa.it (A.-C.M.)
2 Institute of Crop Production, Sant'Anna School of Advanced Studies, Piazza Martiri della Libertà 33, 56127 Pisa, Italy; gaia.monteforti@santannapisa.it
* Correspondence: matteo.dellapiana@santannapisa.it

**Abstract:** Agricultural land abandonment is an increasing concern in the EU, especially in Mediterranean regions where traditional perennial crops like olive groves are left unmanaged. This study focuses on the impact of land abandonment on soil-dwelling arthropods in olive groves in Monte Pisano, Tuscany, examining ants, spiders, myriapods, and carabids. Using Generalized Linear Mixed Models, the potential olive fruit fly predator community was analyzed over two sampling periods repeated over two years to assess the effects of both abandonment and its proximity to managed fields. Ants were significantly more abundant in managed fields next to abandoned ones, though there were no differences between managed and abandoned fields. Spider abundance did not respond to abandonment nor proximity. Myriapods were more abundant in abandoned fields during the first sampling period, but the proximity of an abandoned field had no effect. Carabids were more abundant in managed fields, especially those adjacent to other managed fields. These results indicate that arthropod responses to abandonment are taxon-specific, highlighting that a mosaic of managed and abandoned fields can both enhance and reduce olive fruit fly predator abundance. Conservation strategies should integrate varying management intensities to optimize biodiversity in Mediterranean agroecosystems. Future research should investigate long-term effects and specific predator responses to abandonment.

**Keywords:** predators; spiders; ants; carabids; myriapods; biological control; *Bactrocera oleae* pupae; land use change

## 1. Introduction

Agricultural land abandonment is a growing concern in the EU, with around 30% of agricultural areas under moderate risk and an estimated 2.9% (5 million hectares) projected to be abandoned by 2030. This trend is driven by a complex interplay of bio-physical, structural, market, and policy factors, posing significant environmental and socio-economic challenges [1]. While agricultural intensification in annual cropping systems generally leads to habitat loss and threatens biodiversity [2], land abandonment in perennial cropping systems has produced contrasting effects on biodiversity [3,4]. The intermediate disturbance hypothesis suggests that biodiversity peaks at intermediate levels of disturbance, with both minimally and highly disturbed systems supporting lower biodiversity [5,6]. On the other hand, the intermediate landscape complexity hypothesis [7] suggests that landscapes with intermediate levels of complexity, characterized by a mixture of semi-natural habitats and agricultural areas, tend to support higher biodiversity. This is because these landscapes provide a variety of habitats and resources, facilitating niche differentiation and species coexistence [8]. Land abandonment represents a reduction in the disturbance, and it brings about consequences on landscape composition and configuration by means of modifying the habitat distribution within the landscape, which in turn has a direct effect

on biodiversity [9]. In perennial systems, semi-natural habitats within the landscape play a crucial role in maintaining biodiversity by offering food resources and refuges for various arthropod species [10,11], some of which may prove useful in pest control. It has been demonstrated that both the spatial configuration of semi-natural habitats (SNHs) and their structure affect their attractiveness for beneficial organisms and species of conservation interest [12]. Land abandonment initiates a succession from cultivated land to a semi-natural habitat, but the question remains if this transition is a benefit for biodiversity conservation and for ecosystem services, or if the succession that is especially initially characterized by few dominant and competitive plant species supports more pest species than beneficial ones. In this regard, few studies [13] have been designed to assess this aspect. It has been demonstrated that extensively managed Alpine grasslands support true bug and syrphid abundance, while in the abandoned grasslands, more unique true bug and syrphid species were found [14]. However, the study focused on grasslands that were only recently abandoned or that were managed extensively to avoid the succession towards woodland.

The presence and characteristics of SNHs are vastly affected by land abandonment, which favors the turnover of vegetation [15,16]. In the initial stages of abandonment, plant diversity reaches its peak with herbaceous plants and shrubs coexisting, creating habitats with high vegetation complexity [17]. This can lead to an increase in arthropod biodiversity [18]. In the latter stages of land abandonment, scrub and tree species of the surrounding vegetation become dominant, excluding open-habitat species, and reduce heterogeneity at a landscape scale, resulting in reduced biodiversity levels [19]. This conceptual framework leads to a highly taxon-dependent response to land abandonment [20]. Different taxa exhibit varying responses due to their specific habitat requirements. Some taxa thrive in the early stages of abandonment due to increased plant diversity and habitat complexity [3], while others may benefit from the later stages as succession progresses and habitats stabilize. More work is necessary to unravel the effects of farmland abandonment on arthropod communities, especially in permanent crops, which are underrepresented in the literature, and more specifically in the Mediterranean context which is characterized by its unique climate [21], rich biodiversity [22], and peculiar agricultural system [23]. Here, traditional perennial crops such as olive orchards, a hallmark of Mediterranean agriculture, are increasingly being abandoned due to economic pressures and challenging agricultural practices [24]. One of the main drivers of land abandonment in the olive growing context in these low-input systems in the hilly areas is the damage associated with the olive fruit fly, *Bactrocera oleae* Rossi, which is the most relevant olive pest [25]. The olive fruit fly produces between three and five generations per year in the Mediterranean area, starting in early spring [26]. The adult flies oviposit in olives, and after completing their development, the larvae of the last generation leave the fruits to pupate in the ground, where they are exposed to generalist soil-dwelling predators, such as spiders, carabids, myriapods, staphylinids, and ants [27–30]. Previous works investigated the role of tillage intensity on soil-dwelling predator activity and pupa emergence, highlighting contrasting results. While Bachouche and colleagues [31] found that pupa emergence decreased with increased burial depth, other studies [32,33] observed that increasing tillage reduced predation on olive fruit fly pupae. However, since soil tillage is almost impossible in terraced landscapes, this pest management tool cannot be applied and, moreover, in hilly areas, this would increase soil erosion. For this reason, it is of the utmost importance to boost the activity of natural predators for the control of the olive fruit fly. Despite the importance of olive groves in the Mediterranean region, few studies have tried to evaluate the effect of land abandonment of olive orchards on the associated arthropod functional biodiversity [20,34–36], and to our knowledge, no studies have evaluated the effect of olive grove abandonment on the beneficial soil-dwelling arthropod populations in managed fields flanked by abandoned fields. Recently abandoned orchards still produce olive fruits, which, however, are not harvested and could potentially represent a sink for *B. oleae* oviposition even when they fall to the ground [26]. This could lead to the residual presence of pupae in the soil of recently abandoned fields which, in turn, drives up predators' abundance. These predators could

then move to close-by managed fields where prey availability is even higher because of the higher olive fruit production and therefore increase total predators' abundance in productive fields. Alternatively, the development of dense vegetation in the abandoned groves could hinder the movement of ground-dwelling predators and diminish their presence and activity in abandoned fields.

The aim of this study is to determine (i) the effect of recent land abandonment on the abundance of soil-dwelling predators in traditional olive groves in a Mediterranean context, focusing specifically on those that are potential predators of the olive fruit fly *Bactrocera oleae* at the pupal stage [27,29], such as spiders (*Arachnida: Araneae*), earwigs (*Dermaptera*), carabids (*Coleoptera: Carabidae*), ants (*Imenoptera: Formicidae*), and myriapods (*Myriapoda*); and (ii) the differences in predator abundances between managed groves flanked by abandoned fields and those flanked by other managed fields. Considering the intermediate disturbance hypothesis and the intermediate landscape complexity hypothesis, we hypothesized that (i) the abundance of soil-dwelling predators would be higher in managed olive groves compared to abandoned groves due to intermediate levels of disturbance and the presence of a more permeable vegetation structure and due to the higher availability of olive fruits for oviposition; and that (ii) managed fields flanked by abandoned fields would support higher predator abundance compared to those flanked by other managed fields, as the abandoned fields provide increased habitat complexity and plant resources and due to the spill-over effect from abandoned fields, where prey is less abundant than in managed fields.

## 2. Materials and Methods

### 2.1. Study Area

The study area includes the traditional olive groves of the Monte Pisano area, between the Arno and Serchio rivers in the province of Pisa in Tuscany, central Italy. The studied orchards were spread across three villages: Calci, Vicopisano, and Buti (Figure 1).

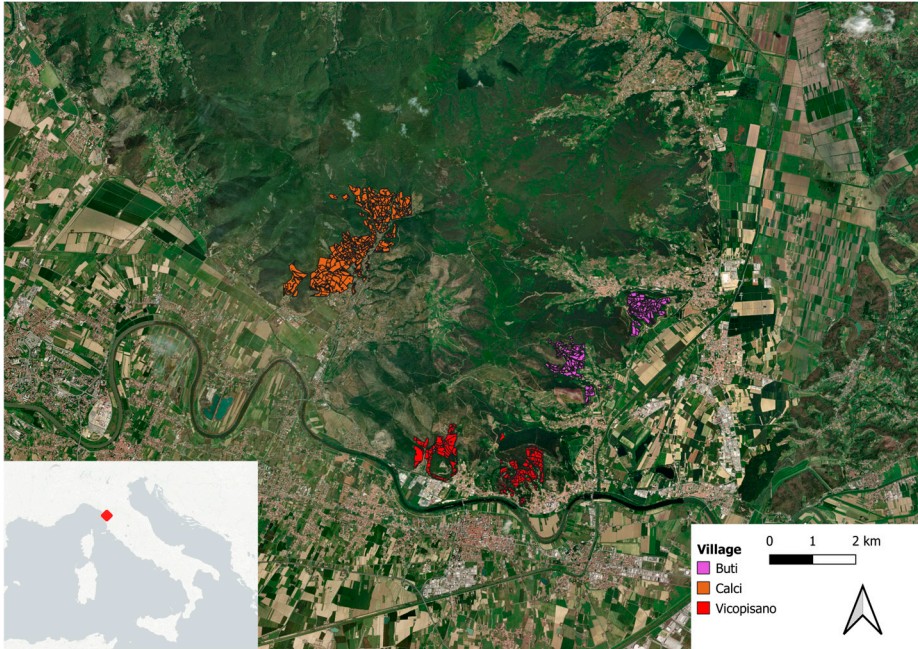

**Figure 1.** Inset map: location of the study area in Central-Western Italy. Main canvas: sampling municipalities in the Monte Pisano area.

Olive growing has strongly characterized the region since the XVII century [37], and most orchards in the Monte Pisano area feature ancient terraces built with dry stone walls. These terraces help manage steep slopes, prevent soil erosion, and improve water retention [11]. Smallholder non-professional farming is the most widespread form of

farming in this area. Generally, farming activities that occupy an area inferior to 10 ha are defined as small-scale farming [38], which is the case for most agricultural activities in the Monte Pisano area.

The climate of the region is a typical hot summer and temperate winter Mediterranean climate, with an average annual rainfall of 1341.7 mm between 2013 and 2023 (http://www.sir.toscana.it/, URL accessed on 11 July 2024, Monte Serra weather station; SD = 231.6 mm), mostly concentrated in autumn, winter, and spring, and the average annual temperature is 14.6 °C. In the area, an increase in extreme rainfall events has been reported [39]; thus, the function of terraces as a drainage system has an increased importance with respect to the past. In the two years of sampling, precipitation was lower than average in 2022 (984 mm, 1.54 SD below the mean), while in 2023 it was more typical (1314.6 mm, 0.12 SD below the mean) (Figure 2).

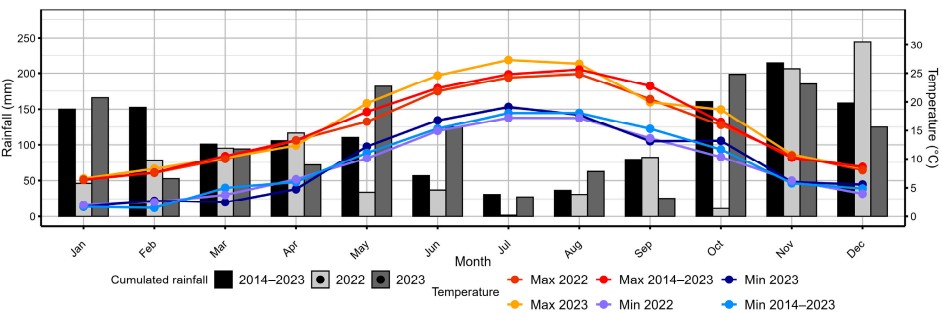

**Figure 2.** Monthly cumulated rainfall and average minimum and maximum monthly temperature, from January 2022 to December 2023.

### 2.2. Sampling Procedure

Five sampling squares of 1 km$^2$ located in the Monte Pisano area were chosen for the experiment, each of them composed of two pairs of fields 500 m apart. One pair consisted of two managed groves (MM), while the other pair was composed of a managed grove and an adjacent abandoned one that still produced olive fruits in the year of sampling (MA). However, the amount of fruit production was very low. The paired groves were facing each other. All managed fields had been cultivated according to organic guidelines for at least 5 years, and no fertilizers had been used in that time lapse. Furthermore, all managed fields were chosen considering their similarity in terms of vegetation cover and mowing pattern. They all had a cover of spontaneous vegetation that was mown once a year, late in the season, mainly to prevent wildfire and to ease the harvesting that takes place by placing nets underneath the trees. The nets can only be managed if the vegetation is cut low. Traps were deployed in two sampling rounds each year over a period of two years, before (T1 = 19 September 2022 and 21 September 2023 installation; 3 October 2022 and 5 October 2023 removal) and after (T2 = 21 November 2022 and 21 November 2023 installation; 5 December 2022 and 5 December 2023 removal) the peak of *B. oleae* pupae in the ground, following the trend reported by Albertini et al. [29]. For each sampling period, three pitfall traps per field were positioned 10 m apart along a transect, totaling 60 traps in T1 and 60 in T2. Traps were plastic vessels (diameter 85 mm; depth 120 mm) filled with acetic acid [60 g$^{-1}$] and saturated with NaCl. Each trap was covered with a plastic roof (height from the ground 20 mm) to prevent flooding and non-target capture and protected with a metallic cage (mesh size 60 mm) to avoid damage caused by wild animals. Traps remained active for 14 days, then samples were recovered and transferred in laboratory for the cleaning procedure, and individuals were stored in ethanol (80%). Quantitative assessments were performed on populations of four taxonomically distinct groups of soil-dwelling predators. Individuals were enumerated from the following groups: *Formicidae* (ants, hereafter), *Carabidae* (carabids), *Myriapoda* (myriapods), and *Araneae* (spiders).

*2.3. Data Analysis*

We analyzed the effect of field management (managed vs. abandoned) and the effect of the proximity to a managed or abandoned field on the soil-dwelling community using Generalized Linear Mixed Models (GLMMs) with a negative binomial distribution to handle overdispersion in the count data. Two separate model sets were constructed: one evaluating the management level (managed vs. abandoned) effect across all fields and another assessing the impact of abandonment proximity within managed fields only. Each set of models was initiated by incorporating fixed effects for sampling time and year, to capture seasonal and annual trends, along with the primary factors of interest: field management status in the first model set and field proximity in the second. Both models also included random effects to account for spatial clustering within the fields, with the individual trap ID nested within the sampling square. Model refinement involved an iterative process where non-significant predictors and interactions were systematically removed. This simplification was guided by likelihood ratio tests (Type II Wald Chi-squared test) and the Akaike Information Criterion. Once the models were simplified to include only significant predictors, final diagnostics were conducted to ensure the validity of the model assumptions. These diagnostics focused on evaluating the residuals for patterns indicating potential issues like non-normality or heteroscedasticity, and for model singularity. Estimated Marginal Means (EMMs) were calculated specifically for the key variables of interest, namely, management level and proximity. The analysis was performed using R Statistical Software (version 4.3.1) [40], making use of the packages glmmTMB (version 1.1.10) [41], DHARMa (version 0.4.7) [42], and emmeans (version 1.10.5) [43]. Plots and figures were created making use of the ggplot2 (version 3.5.1) package [44].

**3. Results**

*3.1. Ants*

Ants were the most abundant group of soil-dwelling predators captured by our pit-fall traps across the two years and the two months both in managed (mean total per field (considering all three traps and both years and sampling times) = 33.68; SD = 52.12) and abandoned (mean = 34.20; SD = 47.00) fields. The analysis, performed with GLMMs (Table 1) aimed at investigating the role of the management level (Figure 3) on ants' abundance, indicated no significant difference between the mean abundances in managed and abandoned fields. The only significant predictor in the model was the time of sampling (Type II Wald test Chi-Squared = 295.92; $p = 2.20 \times 10^{-16}$). The observed ants were 12.79 times more abundant in the first sampling round than in the second round, after the peak of olive fruit fly larvae falling onto the soil. Ants were also the most abundant group in managed fields flanked by another managed field (mean = 28.27; SD = 43.36) and in managed fields flanked by an abandoned field (mean = 44.14; SD = 65.01). The corresponding GLMM (Table 1; Figure 4) highlighted a significant effect of the time of sampling (Type II Wald test Chi-Squared = 261.46; $p = 2.00 \times 10^{-16}$), with an 11.84-fold higher count in the first sampling round. There was also a near-significant effect of proximity, with ants being more abundant in fields flanked by abandoned fields compared to those flanked by managed fields (EMM ratio = 0.68; $p = 0.068$).

**Table 1.** Model parameters for the comparison between management regimes (managed vs. abandoned fields) and proximity (management-flanked vs. abandonment-flanked managed fields) for spiders, ants, carabids, and myriapods. Fixed effects include intercepts, time points (T2), years (2023), and significant predictor and interaction terms. Dispersion parameters and random effects are also reported. Estimates are presented with 95% confidence intervals (CIs) and *p*-values.

| Group | Parameter | Managed Fields vs. Abandoned Fields | | Management-Flanked vs. Abandonment-Flanked Managed Fields | |
|---|---|---|---|---|---|
| | | Estimate (CI) | *p* | Estimate (CI) | *p* |
| | | Fixed Effects | | | |
| Spiders | (Intercept) | 8.34 (7.07, 9.83) | <0.001 | 8.78 (7.21, 10.69) | <0.001 |
| | Time (T2) | 0.23 (0.17, 0.31) | <0.001 | 0.23 (0.17, 0.32) | <0.001 |
| | Year (2023) | 0.67 (0.52, 0.85) | 0.001 | 0.64 (0.48, 0.86) | 0.003 |
| | Time (T2) × Year (2023) | 2.37 (1.59, 3.53) | <0.001 | 2.40 (1.50, 3.85) | <0.001 |
| | | Dispersion | | | |
| | (Intercept) | 3.38 (2.40, 4.77) | | 2.94 (2.03, 4.27) | |
| | | Fixed Effects | | | |
| | (Intercept) | 68.89 (48.08, 98.72) | <0.001 | 59.23 (35.40, 99.11) | <0.001 |
| | Time (T2) | 0.05 (0.04, 0.07) | <0.001 | 0.06 (0.04, 0.09) | <0.001 |
| | Proximity (MA) | | | 1.48 (0.97, 2.25) | 0.068 |
| | | Dispersion | | | |
| Ants | (Intercept) | 0.82 (0.67, 1.01) | | 0.80 (0.64, 1.00) | |
| | | Random Effects | | | |
| | SD (Intercept: Quadrant) | 0.26 (0.07, 1.04) | | 0.48 (0.22, 1.03) | |
| | SD (Intercept: Trap: Quadrant) | 0.35 (0.17, 0.73) | | | |
| | | Fixed Effects | | | |
| | (Intercept) | 1.90 (1.27, 2.84) | 0.002 | 2.06 (1.30, 3.27) | 0.002 |
| | Time (T2) | 0.34 (0.20, 0.55) | <0.001 | 0.35 (0.22, 0.55) | <0.001 |
| | Field management (Abandoned) | 0.20 (0.09, 0.45) | <0.001 | | |
| | Field management (Abandoned) × Time (T2) | 4.10 (1.30, 12.94) | 0.016 | | |
| Carabids | Proximity (MA) | | | 0.62 (0.38, 1.04) | 0.069 |
| | | Dispersion | | | |
| | (Intercept) | 0.62 (0.41, 0.93) | | 0.81 (0.50, 1.31) | |
| | | Random Effects | | | |
| | SD (Intercept: Quadrant) | 0.26 (0.06, 1.03) | | 0.36 (0.14, 0.94) | |
| | SD (Intercept: Trap: Quadrant) | $0.16 \, (3.61 \times 10^{-3}, 6.78)$ | | | |
| | | Fixed Effects | | | |
| | (Intercept) | 3.13 (2.30, 4.25) | <0.001 | 3.13 (2.19, 4.47) | <0.001 |
| | Year (2023) | 1.41 (1.10, 1.81) | 0.007 | 1.38 (1.01, 1.89) | 0.042 |
| | Time (T2) | 0.64 (0.48, 0.86) | 0.003 | 0.65 (0.48, 0.90) | 0.008 |
| | Field management (Abandoned) | 2.01 (1.37, 2.93) | <0.001 | | |
| | Field management (Abandoned) × Time (T2) | 0.55 (0.31, 0.98) | 0.044 | | |
| Myriapods | | Dispersion | | | |
| | (Intercept) | 1.71 (1.27, 2.28) | | 1.40 (1.01, 1.94) | |
| | | Random Effects | | | |
| | SD (Intercept: Quadrant) | 0.23 (0.10, 0.56) | | 0.28 (0.12, 0.68) | |
| | SD (Intercept: Trap: Quadrant) | $3.78 \times 10^{-3}$ (0.00, Inf) | | | |

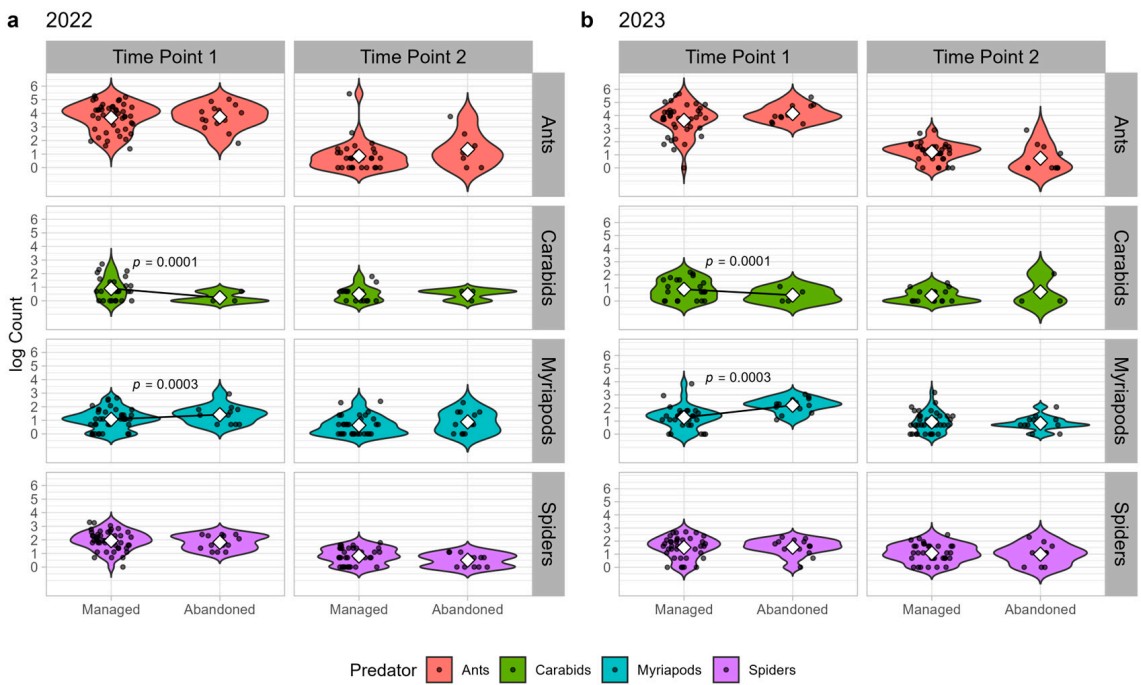

**Figure 3.** Log-transformed counts of soil-dwelling predators (ants, carabids, myriapods, and spiders) in managed and abandoned fields across two time points in 2022 (**a**) and 2023 (**b**). Black points represent individual fields (3 traps aggregated), and (when present) significant differences between means (white squares) are indicated with lines and corresponding *p*-values (adjusted using Tukey's method).

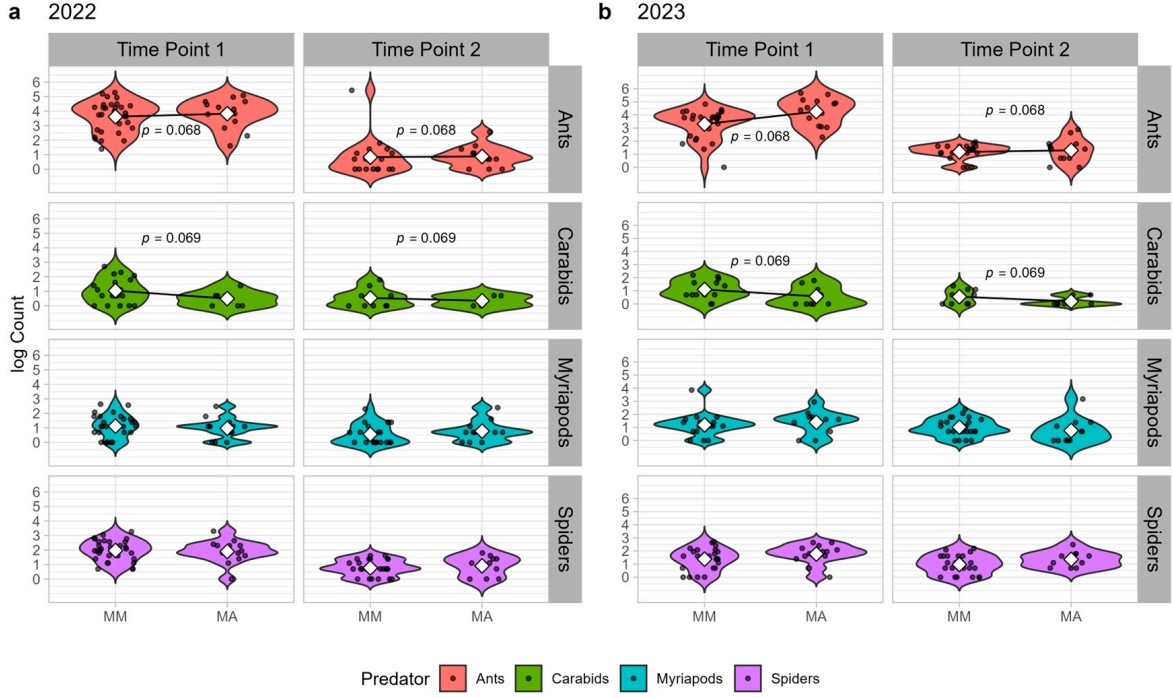

**Figure 4.** Log-transformed counts of soil-dwelling predators (ants, carabids, myriapods, and spiders) in managed fields close to abandoned fields (MA) and managed fields close to managed fields (MM) across two time points in 2022 (**a**) and 2023 (**b**). Black points represent individual fields (3 traps aggregated), and (when present) significant differences between means (white squares) are indicated with lines and corresponding *p*-values (adjusted using Tukey's method).

*3.2. Spiders*

Spiders were the second most abundant group in managed fields (mean total per field (considering all three traps and both years and sampling times) = 4.93; SD = 4.76) and the third most abundant group in abandoned fields (mean = 4.16; SD = 3.29). In managed fields, spiders were the second most abundant group both in the fields flanked by a managed field (mean = 4.85; SD = 4.71) and in the fields flanked by an abandoned field (mean = 5.09; SD = 4.86). The GLMM that focused on the role of the management level (Figure 3) indicated a significant effect of the time of sampling (Table 1) (Type II Wald test Chi-Squared = 102.05; $p = 2.20 \times 10^{-16}$), with an observed 2.84-fold higher count in the first sampling round, and the interaction between the sampling round and the year of sampling (Type II Wald test Chi-Squared = 18.14; $p = 2.05 \times 10^{-5}$). The second model focused on managed fields only and highlighted no significant differences between managed fields flanked by either an abandoned field or by another managed fields (Table 1; Figure 4), again indicating an effect of the time of sampling (Type II Wald test Chi-Squared = 72.61; $p = 2.20 \times 10^{-16}$), with an observed 2.84-fold higher count in the first sampling round, and the interaction between sampling round and year (Type II Wald test Chi-Squared = 13.29; $p = 2.67 \times 10^{-4}$).

*3.3. Myriapods*

Myriapods were the third most abundant group in managed fields (mean total per field (considering all three traps and both years and sampling times) = 3.16; SD = 4.74) and the second most abundant group in abandoned fields (mean = 5.13; SD = 4.82). Restricting the count to managed fields, myriapods were the third most abundant group both in management-flanked (mean = 3.13; SD = 4.98) and abandonment-flanked fields (mean = 3.21; SD = 4.32). The first GLMM, aimed at the comparison of managed and abandoned fields, highlighted a significant effect of the management regime (Table 1; Figure 3) (Type II Wald test Chi-Squared = 9.07; $p = 2.60 \times 10^{-3}$), the year of sampling (Type II Wald test Chi-Squared = 7.30; $p = 6.88 \times 10^{-3}$), the sampling round (Type II Wald test Chi-Squared = 21.86; $p = 2.93 \times 10^{-6}$), and the interaction between management level and sampling round (Type II Wald test Chi-Squared = 4.07; $p = 0.44$). Myriapods were 1.57 times more abundant in the first sampling round than in the second, and 1.47 times more abundant in 2022 than in 2023. The EMMs for this model indicated that the abundance of myriapods was significantly lower in managed fields compared to abandoned fields during the first sampling round in both years, with an EMM ratio of 0.50 ($p = 3.00 \times 10^{-4}$) in 2022 and 2023 at T1, and an EMM ratio of 0.90 ($p = 0.62$) at T2 in both years. The second model, investigating the effect of the proximity to either managed or abandoned fields (Figure 4), showed no significant difference between the two levels (Table 1), only highlighting a significant effect of sampling time (Type II Wald test Chi-Squared = 7.01; $p = 8.09 \times 10^{-3}$) and year (Type II Wald test Chi-Squared = 4.13; $p = 0.04$).

*3.4. Carabids*

The least abundant group was that of carabids, both in managed (mean total per field (considering all three traps and both years and sampling times) = 1.32; SD = 2.24) and abandoned fields (mean = 0.46; SD = 1.24); and in managed fields flanked by another managed field (mean = 1.56; SD = 2.57) or by an abandoned one (mean = 0.84; SD = 1.27). The model comparing managed and abandoned fields highlighted a significant effect of the management regime (Table 1) (Type II Wald test Chi-Squared = 9.00; $p = 2.70 \times 10^{-3}$), sampling round (Type II Wald test Chi-Squared = 13.18; $p = 2.82 \times 10^{-4}$), and their interaction (Type II Wald test Chi-Squared = 5.80; $p = 0.02$). Carabids were more abundant in managed fields compared to abandoned fields during the first sampling round (EMM ratio = 5.09; $p = 1.00 \times 10^{-4}$) (Figure 3), but this difference was not observed in the second sampling round (EMM ratio = 1.24; $p = 0.60$). The observed carabids were 2.52 times more abundant in the first sampling round than in the second one. The model analyzing managed fields flanked by managed or abandoned ones indicated a significant effect of sampling round

(Table 1; Figure 4) (Type II Wald test Chi-Squared = 20.01; $p = 7.70 \times 10^{-6}$) and an almost significant effect of the proximity to a managed or abandoned field (Type II Wald test Chi-Squared = 3.30; $p = 0.07$). Carabids were 1.6 times more abundant in fields flanked by another managed field during both sampling rounds (EMM ratio = 1.60; $p = 0.07$). The observed carabids were 3 times more abundant in the first sampling round than in the second one.

## 4. Discussion

Our study revealed significant variations in the abundance of some groups of soil-dwelling predators based on management regime and the proximity of managed fields to abandoned fields. Contrary to the consistent decline or increase in biodiversity observed with prolonged abandonment in other studies [3,19], our results highlight a taxon-dependent response to land abandonment. Specifically, ants were more abundant in fields flanked by abandoned fields at both sampling times, although no significant differences were observed between managed and abandoned fields themselves. This partially aligns with findings by Altieri and Schmidt [45], who noted that ant abundance increases with an increase in the structural complexity of the vegetation, and with reduced disturbance. More recent studies [46] highlighted a higher abundance of ants in olive orchards compared to forests in a mixed landscape, a result which partially aligns with those presented in this work. Additionally, ants are known as bioindicators of environmental change, showing initial increases in richness and abundance with progressive land abandonment [47]. Moreover, the negative effects of intensive land use on ant populations and their predation efficiency, as observed by Wilker et al. [48], further emphasizes the importance of less disturbed and more heterogeneous habitats for maintaining ant biodiversity and the ecological services they can provide, such as biological pest control. Ants play an important role in agroecosystems as predators. For instance, ant predation on potato tuberworm larvae declines with increased management intensity, indicating their effectiveness as natural pest control agents in less disturbed environments [45]. Mediterranean ant species have been recently reported as indirect control agents of *Ceratitis capitata* Wiedemann [49], the Mediterranean fruit fly, that belongs to the same family as *B. oleae* and shows a similar behavior to the olive fruit fly. We need to stress that our sampling methodology may not have fully captured the diversity and abundance of ant populations. The use of three pitfall traps per field, positioned 10 m apart, might underrepresent ant communities due to their extensive foraging ranges and social structures. Ant sampling often requires more traps placed at greater distances to adequately represent their populations [50]. However, this sampling method was necessary to be able to compare several very different groups of soil-dwelling predators around the optimum time for the predation of the olive fruit fly larvae.

Our study did not observe significant differences in spider abundance between managed and abandoned fields or between managed fields flanked by abandoned fields. The variability in spider responses to different habitat structures and management intensities [51] might explain the lack of significant differences observed in our study. Batáry et al. [52] found that reduced management intensity and increased non-crop habitat, such as field edges, can enhance spider species richness and abundance. This suggests that while our study did not show significant differences, the presence of less disturbed habitats may still play a crucial role in supporting spider populations [53–55]. Additionally, de Paz et al. [20] highlighted that different spider families have varied preferences for habitat structures, with some favoring more complex vegetation found in abandoned groves and others thriving in more open habitats, a trend that has also been observed in mountain pastures [17] and forests [56]. Picchi et al. [10] showed that the *Linyphiidae* family was the only group affected by landscape composition. The lower resolution of the identification performed on spiders in this study may be masking these family-specific trends. Myriapods were significantly more abundant in abandoned fields during the first sampling period but not in the second. Generally, millipedes thrive in environments with high organic matter

and leaf litter, which are more prevalent in less disturbed, abandoned fields [57]. This aligns with the concept that abandoned fields, with their accumulated decomposing plant material and less intensive management, provide ideal conditions for millipede populations. Centipedes, on the other hand, are influenced by habitat complexity rather than the availability of decomposing matter [58]. Accordingly, both groups may benefit from the characteristics found in abandoned fields.

Our study found that carabids were more abundant in managed fields during the first sampling period but not in the second. Carabids also thrived more in managed fields flanked by other managed fields compared to those adjacent to abandoned fields. Carabid beetles prefer habitats with a low vegetation and specific microhabitats for overwintering, such as tussock-forming grasses which provide dry conditions [59]. Managed olive groves offer these conditions, resulting in higher carabid abundance. Accordingly, Do et al. [60] found that carabid species richness and abundance decreased in abandoned fields, likely due to increased vegetation complexity and reduced disturbance, two habitat characteristics that are not preferred by carabids. In Mediterranean olive agroecosystems, maintaining orchard management supports diverse carabid populations, which enhances pest control services [61]. Our sampling periods, limited to October and November, surely have not captured the full pattern of carabid community development. Carabids exhibit complex seasonal dynamics, with activity peaks that vary throughout the year [59]. Repeated sampling across multiple seasons is recommended to obtain a comprehensive understanding of their population trends and responses to land management practices [61]. However, as mentioned above, the interest of this study was the relative comparison of carabid abundance patterns as a response to olive grove abandonment.

Our results suggest that the intermediate disturbance hypothesis may not fully explain the patterns observed in perennial cropping systems like olive groves. The increased abundance of certain predators in abandoned fields suggests that these areas provide essential habitats and resources that support predator communities, though this effect varies among taxa. Furthermore, we only took into account groves that were abandoned recently, between 2 and 10 years ago, because these groves continue to produce some olives. Long-term abandonment is likely to have a different effect on the predator community since the supply of olive fruit fly pupae will be interrupted when trees stop producing olives. Regarding the intermediate landscape complexity hypothesis, our findings indicate that the presence of abandoned fields close to managed ones can enhance biodiversity in managed fields by providing diverse habitats and resources, but in the case of carabids, the proximity of abandoned fields can lower the presence of natural predators in managed fields.

One limitation of our study is its relatively short duration, which may not capture long-term trends in predator populations. Additionally, focusing on a specific region in Tuscany limits the generalizability of our findings to other Mediterranean landscapes. Furthermore, relying solely on abundance as a measure surely does not provide insight into the species richness of the studied groups. However, this dimension goes beyond the scope of this paper. Despite these limitations, our results indicate that the mosaic of managed and abandoned fields could affect functional biodiversity by providing diverse habitats and resources for soil-dwelling predators, although the direction of the effect can be both positive and negative, depending on the predator group. This supports the idea that land abandonment can be tolerable or even beneficial up to a certain point, with a gradient of management levels being optimal for landscape biodiversity. Abandonment in our study was relatively young because fields were selected with trees that still produced at least some olives. It is quite frequent in the studied region that these olive groves go through a cycle from low-intensity management to short abandonment [62], and then back to managed fields again. These dynamics depend on social-economic processes connected to the age of the owners, the profitability of olive oil production, and the selling or renting of olive groves to new owners. Studies have shown that a mix of habitats, ranging from highly managed to semi-natural, supports a greater variety of species and ecological functions [34–36,63]. Therefore, biodiversity conservation strategies should consider both

farm- and field-scale agricultural practices and landscape configuration [24]. Future research should explore the long-term effects of land abandonment on predator populations and extend the study to different Mediterranean regions. Investigating the interactions between different predator taxa and their prey could provide deeper insights into the ecological dynamics of abandoned and managed fields. Additionally, studies should focus on the landscape features that influence these dynamics, such as land use composition, as highlighted by recent research [64].

## 5. Conclusions

Our study investigated the dynamics of soil-dwelling predators in a Mediterranean olive grove agroecosystem affected by land abandonment. By comparing managed and abandoned fields, we observed that taxa such as myriapods tend to thrive in abandoned groves, likely due to increased vegetation complexity and reduced disturbance. In contrast, carabids were more abundant in managed fields, particularly those adjacent to other managed areas, confirming that beetles prefer the specific habitat conditions provided by low-intensity agricultural practices. Conversely, ants showed a preference for managed fields flanked by abandoned areas, reflecting their adaptability and role as bioindicators of environmental change. The absence of significant differences in spider abundance across different field types highlights the variability in species responses to habitat changes. These findings suggest that maintaining a mosaic landscape with fields characterized by very low-to-intermediate management intensity can be important for promoting ecosystem services such as pest predation. The transition from management to abandonment should, however, be monitored and regulated by local stakeholders, with the aim of providing no harm to active farms and no damage to the agroecosystem. Future research should focus on understanding the long-term implications of land abandonment, particularly how landscape configuration and habitat features influence predator–prey dynamics and overall agroecosystem health in Mediterranean regions.

**Supplementary Materials:** The following supporting information can be downloaded at: https://www.mdpi.com/article/10.3390/land13111845/s1, Data Matrix: pitfall traps sampling dataset.

**Author Contributions:** Conceptualization: M.D., A.C., R.P. and A.-C.M.; methodology: M.D., A.C., G.M. and R.P.; data curation: A.C. and G.M.; data analysis and writing—original draft preparation: M.D.; writing—review and editing: A.C. and A.-C.M.; funding acquisition: A.-C.M. All authors have read and agreed to the published version of the manuscript.

**Funding:** This study is part of the FRAMEwork project funded by the European Union's Horizon 2020 research and innovation program under grant agreement No 862731.

**Data Availability Statement:** The data presented in this study are available as Supplementary Materials (Supplementary Data Matrix).

**Acknowledgments:** The authors are grateful to Malayka Samantha Picchi for the assistance provided during the writing process. The authors are grateful to the olive farmers for allowing us to sample in their olive groves.

**Conflicts of Interest:** The authors declare no conflicts of interest.

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
