# Peer review of "Soil-Dwelling Arthropods’ Response to Land Abandonment Is Taxon-Specific in a Mediterranean Olive Grove Agroecosystem"

_land, doi:10.3390/land13111845_

Round 1
Reviewer 1 Report
Comments and Suggestions for Authors
Comments to "Soil-Dwelling Arthropods' Response to Land Abandonment is Taxon-Specific ..." by Matteo Dellapiana et al.
The Authors investigate how land abandonment might affect the abundance of different soil arthropods in olive-groves, namely ants, spiders, myriapods, and carabids).
The main finding is that there is no unique response common to all groups and that the reaction of each group should be carefully considered by any plan to optimize biodiversity conservation in agroecosystems.
I think the paper is interesting, well designed and presented, although the text needs some polishing here and there.
My main concern is that the sampling scheme adopted may somewhat underrepresent the response of some of these groups. I think this point should be carefully addressed in the discussion.
1) For each sampling period, three pitfall traps per field were
positioned 10 meters apart along a transect. Such a sampling scheme may be useful for some of the taxa (e.g. Carabids) but not others, like ants, for which more traps placed at a distance of at least 25 meters are recommended (there is plenty of literature on this, even in olive orchards).
Conversely, despite two sampling periods may be sufficient to get a clear representation of an ant community (although October/November is not the peak of their activity), it is not optimum for Carabids, which have complex phenologies, and repeated samples throughout the year are recommended.
2) Abundance is not itself a really useful measure of biodiversity for ants, as they can form colonies of thousands of individuals. Using simple abundance without any control of species richness and diversity may lead to considering a site dominated by a single, highly abundant species (even invasive) more diverse than a site with many species with smaller colonies. The problem of sample collection raised above may further exacerbate this effect.
I synthesis I think this paper can be accepted for publication, but the authors should address in the discussion section the limitations of the sampling scheme and analysis for each of the considered taxa.
Below I add a list of minor points
line 50: The acronym SNH must be defined at first use.
lines 52-56: This sentence is too long and hard to read please rephrase. Also please check and correct "beneficias".
line 57: move the quotation to the end of the sentence.
lines 68-70: This sentence is awkwardly written; please rephrase.
Comments on the Quality of English LanguageThe language is generally acceptable, although some sentences could be rephrased to improve clarity
Author Response
Comment 1: “For each sampling period, three pitfall traps per field were positioned 10 meters apart along a transect. Such a sampling scheme may be useful for some of the taxa (e.g. Carabids) but not others, like ants, for which more traps placed at a distance of at least 25 meters are recommended (there is plenty of literature on this, even in olive orchards). Conversely, despite two sampling periods may be sufficient to get a clear representation of an ant community (although October/November is not the peak of their activity), it is not optimum for Carabids, which have complex phenologies, and repeated samples throughout the year are recommended.”
Response 1: We thank the Reviewer for the comment and agree on the constraints of the sampling method we deployed in the present study. In accordance with the suggestion, a further explanation of the limitations of the study has been included in the Discussion section's paragraphs regarding ants (lines 317-321) and carabids (lines 355-360), specifically referring to the suboptimal sampling design. The added explanation underlines the possible underrepresentation of the ant population and the temporal constraints of the carabids’ sampling
Comment 2: “Abundance is not itself a really useful measure of biodiversity for ants, as they can form colonies of thousands of individuals. Using simple abundance without any control of species richness and diversity may lead to considering a site dominated by a single, highly abundant species (even invasive) more diverse than a site with many species with smaller colonies. The problem of sample collection raised above may further exacerbate this effect.”
Response 2: We agree on the observation regarding the problematic consequences of using abundance as a measure of biodiversity, but we stress that the focus of the study is to compare the potential availability of an ecosystem service such as predation across different fields, not comparing biodiversity. In accordance with the suggestion, a short sentence highlighting the problematic nature of abundance has been added to the final part of the Discussion section (lines 377-379), with the aim of clarifying the focus of the presented study on the presence of soil-dwelling predators or lack thereof.
Reviewer 2 Report
Comments and Suggestions for Authors
The study is original and well-defined. It provides advancement and some contribution to the current knowledge in ground-dwelling arthropods. The latter is significant and required element of soil cenosis providing its fruitfulness. The topic of the manuscript fits the aims and scope of Land. The data is accurately interpreted and robust enough to draw conclusion. The methods, tools, softwares are described in sufficient detail to allow the replication of results.
But I have some notes.
1. Pitfall traps are used widely when sampling soil-dwelling arthropods, but ants must be sampled by other methods (nests, ant trials etc.). The latter are described in the classic manuscripts (f. e. Methods of soil zoological studies. Publishing House Nauka, Moscow, 1975). So I recommend deleting data in ants from the MS.
2. Data of the MS is interesting to a wide range of specialists in soil biota. So I highly recommend to include lists of taxones sampled in each area into the MS with designation in Latin.
Author Response
Comment 1: “Pitfall traps are used widely when sampling soil-dwelling arthropods, but ants must be sampled by other methods (nests, ant trials etc.). The latter are described in the classic manuscripts (f. e. Methods of soil zoological studies. Publishing House Nauka, Moscow, 1975). So I recommend deleting data in ants from the MS.”
Response 1: We thank the Reviewer for the observation, regarding the sampling methods for ants and its inherent limitations. We agree that pitfall traps have limitations in capturing the full spectrum of ant diversity and abundance. However, our study aimed to compare multiple groups of soil-dwelling predators using a consistent sampling method across all taxa. By employing pitfall traps, we ensured comparability in the relative abundance patterns observed among different predator groups under varying management regimes. An explanation of the limitation of the study has been added in the paragraph regarding ants in the Discussion section (lines 317-321). The added explanation underlines the possible underrepresentation of the ant population.
Comment 2: "Data of the MS is interesting to a wide range of specialists in soil biota. So I highly recommend to include lists of taxones sampled in each area into the MS with designation in Latin."
Response 2: We agree on the potential interest of providing the species' list, however the present study relied on a very broad taxonomical classification, which is the one presented in the Materials and Methods section at lines (lines 175-178). The aim of this paper is to assess how abundance patterns of several potential predators of the olive fruit fly larvae respond to the presence of abandoned olive groves. We therefore sampled the soil predators at the peak of the olive fruit fly larvae migration from the olives into the soil (T1) and after the peak (T2). In future work we will explore the species composition of some of these groups in more detail. The key message of this paper is that the presence of abandoned groves does not necessarily cause a negative effect on all potential predator groups.